# Results of an international survey on the use and perceptions of the laryngeal mask airway (LMA)

**Natalie Krug** [ID]*, **Volker Thieme, Maria Theresa Voelker** [ID], **Sven Bercker**

Department of Anaesthesiology and Intensive Care Medicine, University Hospital of Leipzig, Leipzig, Saxony, Germany

* Natalie.Krug@medizin.uni-leipzig.de

## Abstract

### Background and objectives

Laryngeal Mask Airway (LMA) are valuable tools for airway management in anaesthesia and offer advantages in various scenarios. However, due to a lack of data, for many aspects of its application no standards and clear limitations are used.

### Methods

Between May 15th and June 15th 2023, a link to an online-based survey was sent via email to members of the European Society of Anaesthesiology and Intensive Care (ESAIC). The survey included 16 questions on attitudes regarding weight limit, patient positioning, positive end-expiratory pressure (PEEP) and peak inspiratory pressure (PIP) limits, surgical procedures, their duration and ventilation modes when using LMA as well as level of experience and nationality.

### Results

The survey was sent to 7145 recipients, 345 of whom completed the survey. The respondents came from 54 countries, including 31 from European countries. 121 anaesthetists (31.9%) reported considering an upper limit of Body-Mass-Index (BMI) when using a LMA (median 35 kg/m²), 186 (49.1%) reported a maximum duration of use (median 120 minutes), 223 anaesthetists (58.8%) reported a PEEP limit (median 5 mBar) and 238 (62.8%) a PIP limit (median 22,5 mBar). 179 (47.2%) use LMA only in supine position, 53 (14%) in supine and prone position and 147 (38.8%) in supine, side and prone position. The majority (n=322; 85%) do not use LMA for procedures with increased intra-abdominal pressure such as laparoscopy.

### Conclusion

The results of this survey demonstrate very heterogeneous practices and perceptions of ESAIC anaesthetists regarding the use of LMA in different circumstances. Our data suggest that there is no consistent or widely accepted strategy on these issues. Anaesthetists

**Data availability statement:** All relevant data are within the manuscript and its Supporting Information files.

**Funding:** The author(s) received no specific funding for this work.

**Competing interests:** The authors have declared that no competing interests exist.

should have a strong interest in resolving these uncertainties by conducting international randomised prospective trials.

## Introduction

Since its introduction in the 1980s [1], the Laryngeal Mask Airway (LMA) has become a cornerstone of airway management in anaesthesia, offering numerous advantages over endotracheal intubation, including a reduced incidence of complications. Nevertheless, despite its proven benefits, significant uncertainties remain regarding its use in complex clinical scenarios. These include obese patients, cases with increased intra-abdominal pressure, operations exceeding two hours, and specific patient positions like the lateral or prone positions. Furthermore, there is a paucity of evidence regarding the optimal ventilation modes and pressures for LMAs, with these aspects remaining largely unexplored.

This dearth of robust evidence contributes to variability in the practices and perceptions of anaesthetists globally. Guidelines by professional societies these primarily address difficult airway management, rather than the specific limitations of LMA use in the aforementioned scenarios [2–5].

The objective of the here presented international survey was to investigate the heterogeneity of LMA usage across diverse clinical environments and scenarios. The identification of existing gaps and regional differences is intended to provide a foundation for multicentre studies that could enhance the evidence base. The survey's findings are expected to underpin the creation of evidence-based guidelines, the harmonisation of standards of care, and the optimisation of LMA use in challenging settings. An international perspective is essential to ensure the safety and efficacy of LMA in complex clinical contexts, thereby fostering innovation and standardisation in airway management.

## Methods

In accordance with the German medical professional code of conduct (Berufsordnung) consultation with an ethics committee is not required for the collection of data that cannot be attributed to a specific person. Therefore, the local ethics committee waived the need for consultation.

A link to an online-based survey was sent by email to all members of the European Society of Anaesthesiology and Intensive Care (ESAIC). As a rule of ESAIC surveys are sent once without the possibility to send a reminder. Between May 15th and June 15th 2023 all ESAIC members were asked to answer 16 questions on weight limit, patient positioning, positive end-expiratory pressure (PEEP) and peak inspiratory pressure (PIP) limits, surgical procedures, their duration, and preferred ventilation modes while using LMA. Further questions addressed the level of experience and nationality of the anaesthetists. The data was collected and stored anonymously using the questionnaire platform Evasys (Evasys GmbH, Lüneburg, Germany). Implausible values were excluded from further analysis.

For comparative statistics we used Goodman-Kruskal-Tau-A test for ordinal data and the Cramérs V test for nominal data.

Subgroup analyses were performed for duration of professional experience as an anaesthetist (<6 years; 6–10 years; 11–15 years; 16–20 years and >20 years) and the regions of origin of the participants. Goodman-Kruskal-Tau and Cramér's V were calculated to analyse the correlation between professional experience and the limitations of using the LMA. Descriptive data are displayed as boxplots and given as median, quartiles and extremes. Quantitative data for upper limits of PIP, PEEP, BMI and duration were only analyzed when a limit was given by the respondents.

## Results

The survey was sent by email to 7145 recipients, of whom 345 completed the survey.

Countries of origin were distributed very heterogeneously. The respondents hailed from 54 countries worldwide, with 31 of these located in Europe, 12 in Africa, 12 in Asia, 4 in America and 2 in Oceania. The detailed data for the countries and regions are listed in Tables 1 and 2.

The median age of the anaesthetists was 47 years (range 27–75), with a median professional experience of 17 years (range 1–50). 258 anaesthetists (68,1%) reported that they do not use an upper limit for Body Mass Index (BMI) when using a LMA; all others reported to respect a median upper BMI limit of 35 kg/m² (35; 40). 186 (49.1%) of all respondents limited the use of LMA for a median duration of 120 minutes (90; 165). 223 (58.8%) of the 345 participants reported an upper PEEP limit (median 5 mBar (5; 8)) and 238 (62.8%) an upper PIP limit (median 22,5 mBar (20; 28,5)) (Fig 1). 351 (92.6%) of the respondents use controlled

**Table 1. Respondents´ countries origin.**

| Country 1 | [n] | Country 2 | [n] | Country 3 | [n] |
|---|---|---|---|---|---|
| Germany | 50 | India | 6 | Bangladesh | 1 |
| Switzerland | 42 | Denmark | 5 | Mauritius | 1 |
| Portugal | 20 | Hungary | 5 | Bulgaria | 1 |
| Netherlands | 19 | Malta | 5 | Chile | 1 |
| Sweden | 18 | Latvia | 4 | Saudi Arabia | 1 |
| Spain | 18 | Croatia | 4 | Kosovo | 1 |
| United Kingdom, | 17 | Albania | 3 | Catalonia | 1 |
| Belgium | 17 | Estonia | 3 | Finland | 1 |
| France | 11 | New Zealand | 2 | China | 1 |
| Austria | 11 | South Africa | 2 | Moldova | 1 |
| Greece | 11 | Brazil | 2 | Qatar | 1 |
| Poland | 8 | Romania | 2 | Ethiopia | 1 |
| Australia | 8 | Egypt | 2 | Tunisia | 1 |
| Norway | 7 | Serbia | 2 | United Arab Emirates | 1 |
| Czech Republic | 6 | Indonesia | 2 | | |
| Turkey | 6 | Malaysia | 2 | | |
| Slovakia | 6 | Canada | 2 | | |

**Table 2. "Countries/regions".**

| Region | Countries [n] | Participants [n] | Country |
|---|---|---|---|
| Europe | 31 | 313 | Albania, Austria, Belgium, Bulgaria, Catalonia, Croatia, Czech Republic, Denmark, Estonia, Finland, France, Germany, Greece, Hungary, Ireland, Italy, Kosovo, Latvia, Malta, Moldova, Netherlands, Norway, Poland, Portugal, Romania, Slovakia, Slovenia, Spain, Sweden, Switzerland, United Kingdom |
| Africa | 5 | 7 | Egypt, Ethiopia, Mauritius, South Africa, Tunisia |
| Asia | 12 | 30 | Bangladesh, Hong Kong (China), India, Indonesia, Israel, Japan, Malaysia, Qatar, Saudi Arabia, Turkey, United Arab Emirates, Yemen |
| America (North and South) | 4 | 8 | Brazil, Canada, Chile, United States |
| Oceania | 2 | 10 | Australia, New Zealand |

mechanical ventilation (MV) with pressure support ventilation (PSV), 26 (6.9%) use only PSV and 2 (0.5%) only spontaneous breathing without pressure support. 179 (47.2%) anaesthetists use the LMA only in supine position, 53 (14%) use the LMA in supine and prone position and 147 (38.8%) in supine, prone, and lateral position. The range of the lateral position varied between 10° to 180°, with a median of 90°. 57 (15%) reported to use LMA during procedures involving increased intra-abdominal pressure such as laparoscopy. Results are shown in Figs 1–4.

The subgroup analysis between professional experience and restrictions in the use of the LMA did not reveal any significant association for the correlation in the Goodmann Kruskal tau A test and Cramer's V test.

## Discussion

Data presented here show a remarkable heterogeneity in individual standards regarding the use of LMA. We hypothesize that this is based on uncertainty in terms of evidence and that this uncertainty leads to a wide range of individual, local, regional or national standards – or

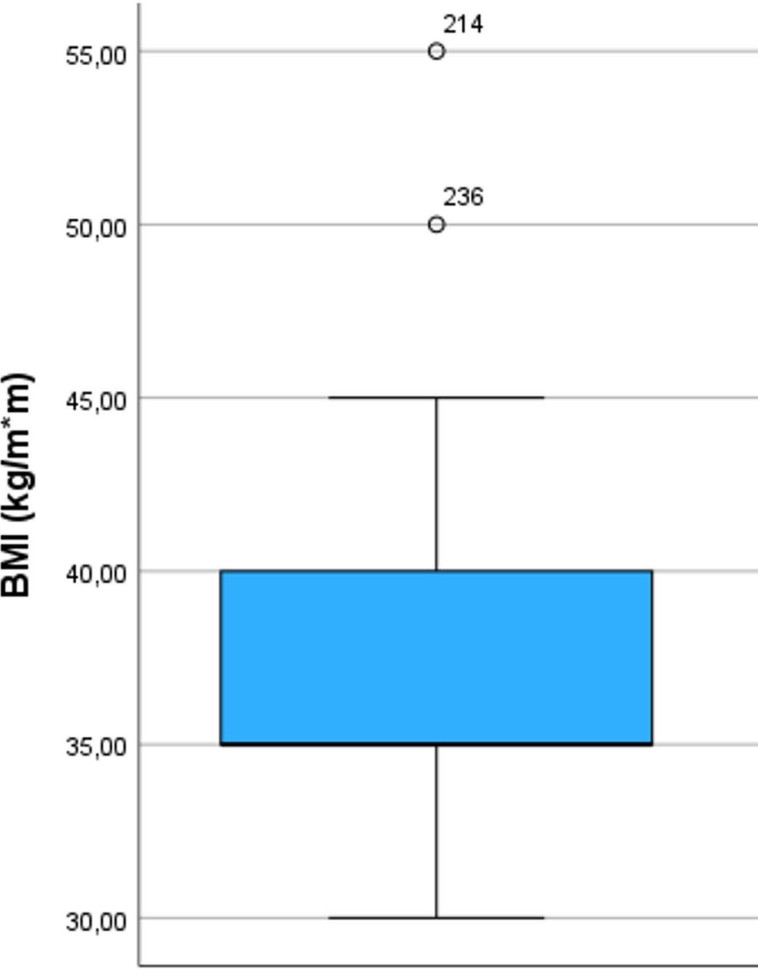

**Fig 1. Distribution of responses regarding asked limitations of BMI.** Boxplots include median, quartiles, extremes and outliers. BMI (Body Mass Index). Data are shown only for respondents who gave an upper limit.

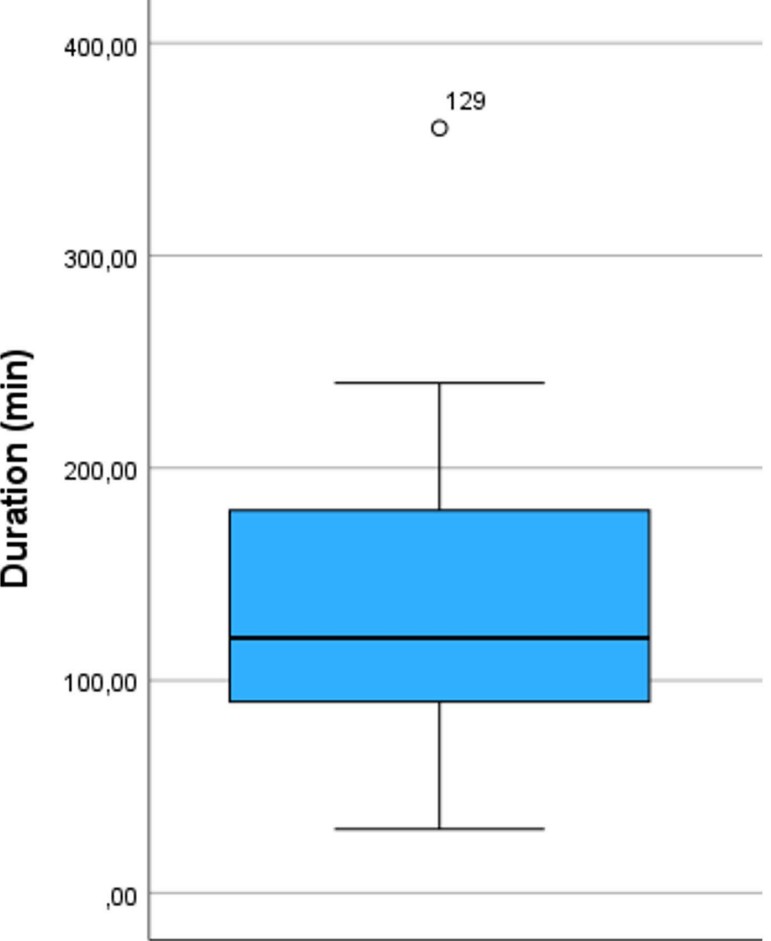

**Fig 2. Distribution of responses regarding asked limitations of duration.** Boxplots include median, quartiles, extremes and outliers. Data are shown only for respondents who gave an upper limit.

lack of thereof. Overall, there is not a very good and, in some respects, not even a satisfactory data basis for many aspects of the intraoperative use of LMA.

## LMA and mechanical ventilation

Positive effects of PEEP during MV in general anaesthesia are well known [6]. However, concerns remain about the use of high PEEP and PIP levels when using LMA. It is assumed that an increase of PEEP while maintaining driving pressure increases the peak inspiratory pressure (PIP) and consequently the incidence of leakage. This assumption may be one reason for the cautious use of PEEP in LMA in our survey. However, prospective data on this assumption are lacking and a recent prospective clinical study could not demonstrate differences in leakage or respiratory function when comparing PEEP versus zero end-expiratory pressure (ZEEP) [7].

A small minority of all respondents use LMA only in spontaneous ventilation mode or in combination with pressure support. It might be postulated that avoidance of any positive pressure ventilation leads to less leakage when using LMA. However, to date, there is no evidence that this strategy affects clinically relevant endpoints.

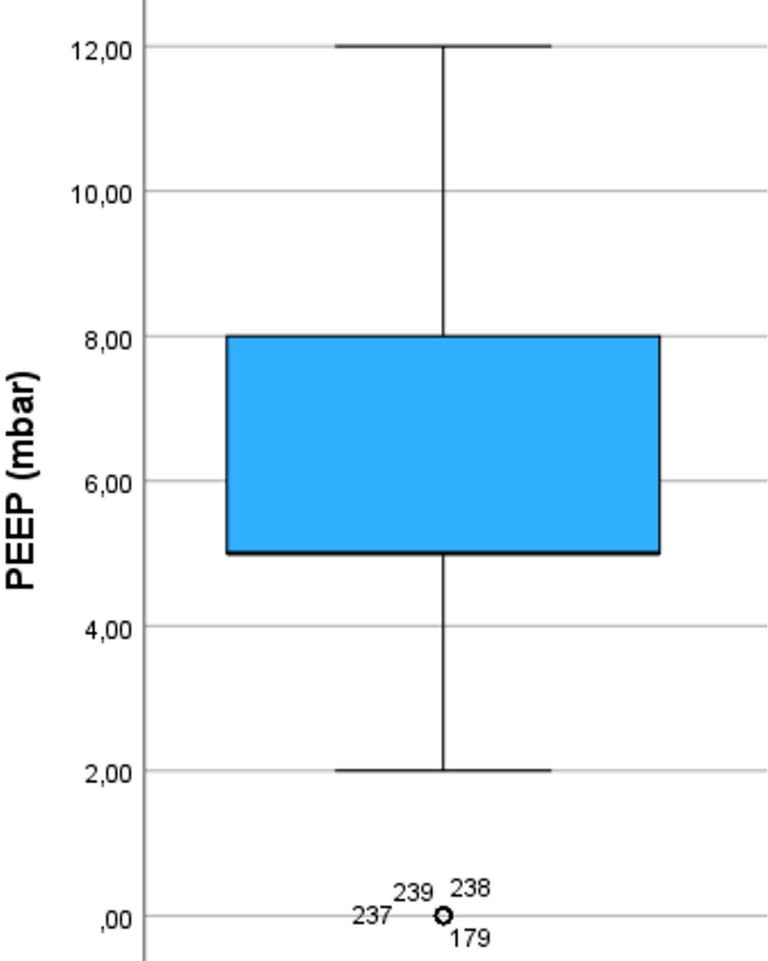

**Fig 3. Distribution of responses regarding asked limitations of PEEP.** Boxplots include median, quartiles, extremes and outliers. PEEP (positive endexpiratory pressure). Data are shown only for respondents who gave an upper limit.

## LMA and procedural duration

This survey did not provide a consistent picture regarding the duration of surgical procedures with LMA use. Half of the respondents (49.1%) specified a time limit, which ranged from 30 to 360 minutes. It is assumed that the risk of regurgitation, aspiration and respiratory insufficiency increases with the duration of surgery. However, there is no clinical evidence supporting this limit. Animal studies suggest that a use for up to eight hours might be safe [8].

## LMA and weight limits

In our survey, the upper weight limits for LMA use varied, with a BMI range from of 30–55 kg/m². The idea of an upper BMI limit for the use of LMAs is based on the pathophysiological assumption that obesity is associated with an increased risk of aspiration, a lower functional residual capacity and requires higher ventilation pressures due to lower lung compliance [9]. These aspects promote higher leakage and inadequate ventilation [10]. Additionally, LMA placement tends to be more difficult in obese patients. However, it is not known up to what BMI the acceptable advantages of a LMA outweigh its disadvantages. Few studies have

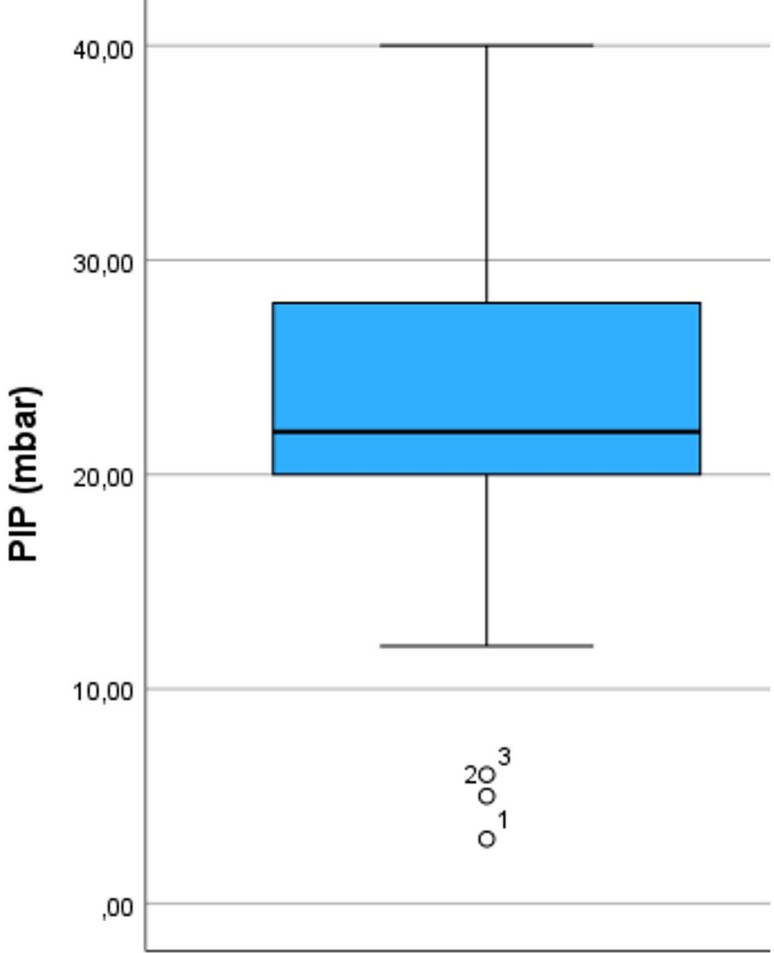

**Fig 4. Distribution of responses regarding asked limitations of PIP.** Boxplots include median, quartiles, extremes and outliers. PIP (peak inspiratory pressure). Data are shown only for respondents who gave an upper limit.

compared endotracheal intubation (ETI) and LMA during elective surgery in obese patients. E.g. Zoremba et al. randomised 134 patients with a BMI of 30–35 kg/m² and demonstrated significantly better lung function including a better $SpO_2$ 24 hours post procedure in the LMA group [11]. In 2013, a Cochrane review on LMA vs. ETI included two RCTs concerning this question [12]. Due to insufficient data these studies did not result in sound recommendations and the authors were unable to draw sufficient conclusions on safety.

## LMA and surgical procedures

Since the early days of LMA use, cases or case series have demonstrated the use in the prone position [13,14]. Advantageous aspects of using the LMA in the prone position, such as the independent positioning of the patient, with less personnel required, a quicker start to the operation, less muscle and joint pain as a surrogate parameter for positioning damage and improved hemodynamic stability [15], have so far only been insufficiently proven. The reports also highlight potential disadvantages of LMA in the prone position, such as the risk of displacement, airway obstruction, and aspiration. The results of our survey, in which about half of the anaesthesiologists rejected prone application, is an expression of this profound lack of evidence.

In contrast, numerous randomized studies and reviews have shown that the use of LMA in laparoscopic surgery could be safe and feasible [16]. In a systematic review Belena et. al could state that the frequency of regurgitation and aspiration associated with the use of the LMA in laparoscopic surgery is very low [17]. In a study by Matlby et. al different LMA and ETI regarding ventilation and gastric distension in normal weight and obese patients during gynaecological laparoscopic surgeries were examined. The LMA (Classic and ProSeal) and ETI appeared to be equally effective in the obesity group (BMI>30 kg/m²) [18]. Already in 2002, Maltby et al. were able to show similar results in laparoscopic cholecystectomies and obese patients (BMI > 30 kg/m²) [18,19]. Accordingly, the current German S1 guideline on airway management (AWMF Registry No.: 001/028) continues to recommend the use of 2nd generation LMA for so-called extended indications, such as laparoscopic procedures, in line with the 2015 guideline [5].

Despite these results, our survey revealed inconsistent areas of application and limitations in the use of LMA during laparoscopic surgery.

## Heterogeneity of LMA use

Our data shows considerable differences in the use of LMA among anaesthetists.

Such variations may arise from differences in healthcare systems, resources and training.

It can be assumed that experienced anaesthetists have more confidence in LMA use and use it for extended indications such as obese patients or prolonged procedures. Inexperienced anaesthetists may act more cautiously, often influenced by theoretical knowledge and safety concerns. However, the respondents' level of professional experience was not significantly associated with the surveyed limitations of LMA use. Therefore, we postulate that differences are primarily attributable to uncertainties and a paucity of evidence. However, this reference to the lack of evidence may be too easy. As already discussed, of all the topics addressed in our survey, the safe use of supraglottic airway devices during laparoscopic procedures is the one for which there is the most scientific evidence. However, this question also shows great heterogeneity and a substantial reluctance to use these devices [16–19]. We can only speculate about the reasons for this but assume that there is insufficient evidence for some questions on the one hand and insufficient acceptance of existing evidence on the other. In short, the traditional strategies from the time when the LMA was developed appear to play a greater role than the available evidence. A major shortcoming in the elective use of LMA is the lack of guidelines. We hypothesise that such a guideline could help to ensure that, in our view, strategies that are no longer up to date are adapted.

## Limitations

An important limitation of this study is the low response rate of less than 5%, which considerably limits the representativeness of the results and may lead to selection bias, as primarily interested or specialised anaesthetists may have participated. Additionally, focusing on ESAIC members may have excluded diverse practices due to differences in healthcare systems, training, and regional standards. The exclusive online format and one-month data collection period might have further limited participation, excluding subgroups. These limitations necessitate caution when interpreting the results and highlight the need for further studies with higher response rates and broader participant bases.

## Conclusion

The results of this survey provide insight into the very heterogenous current practices and perceptions of ESAIC anaesthesiologists regarding the use of LMA under different

circumstances. Our data showed that there seems to be no uniform or widely accepted strategy concerning a variety of topics.

It should be of strong interest to address these uncertainties by conducting randomised prospective studies.

## Supporting information

**S1 File. Data sheet.** xxxx.
(XLSX)

## Acknowledgments

Many thanks to the staff from ESAIC for supporting our survey.

## Author contributions

**Conceptualization:** Natalie Krug.

**Data curation:** Volker Thieme.

**Formal analysis:** Volker Thieme.

**Methodology:** Natalie Krug, Sven Bercker, Volker Thieme.

**Resources:** Sven Bercker.

**Software:** Volker Thieme.

**Supervision:** Maria Theresa Voelker, Sven Bercker.

**Validation:** Natalie Krug, Maria Theresa Voelker, Sven Bercker.

**Writing – original draft:** Natalie Krug.

**Writing – review & editing:** Maria Theresa Voelker, Sven Bercker, Volker Thieme.

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
