## [Decision Letter · Decision Letter 0]

10 Dec 2024

PONE-D-24-43649Results of an international survey on the use and perceptions of the Laryngeal Mask Airway (LMA)PLOS ONE

Dear Dr. Krug,

Thank you for submitting your manuscript to PLOS ONE. After careful consideration, we feel that it has merit but does not fully meet PLOS ONE’s publication criteria as it currently stands. Therefore, we invite you to submit a revised version of the manuscript that addresses the points raised during the review process.

We look forward to receiving your revised manuscript.

Kind regards,

Chinh Quoc Luong, MD., PhD.

Academic Editor

PLOS ONE

Journal Requirements:

Additional Editor Comments:

Thank you for submitting your manuscript to PLOS ONE. After a comprehensive review, the Reviewers have recommended major revisions to your manuscript. We encourage you to address the comments below and resubmit your manuscript for further consideration.

Reviewers' comments:

Reviewer's Responses to Questions

**Comments to the Author**

1. Is the manuscript technically sound, and do the data support the conclusions?

Reviewer #1: Yes

Reviewer #2: Partly

Reviewer #3: Partly

2. Has the statistical analysis been performed appropriately and rigorously? 

Reviewer #1: No

Reviewer #2: Yes

Reviewer #3: No

3. Have the authors made all data underlying the findings in their manuscript fully available?

Reviewer #1: Yes

Reviewer #2: Yes

Reviewer #3: No

4. Is the manuscript presented in an intelligible fashion and written in standard English?

Reviewer #1: Yes

Reviewer #2: Yes

Reviewer #3: No

5. Review Comments to the Author

Reviewer #1: The manuscript presents an overview of an international survey on the use and perceptions of the Laryngeal Mask Airway (LMA) among anesthesiologists, focusing on aspects such as BMI limits, patient positioning, PEEP, PIP, and procedural duration. While the topic is timely and highly relevant to clinical practice, the manuscript requires substantial revisions in several areas to enhance its methodological rigor and practical applicability.

The response rate of the survey was low (345 out of 7,145 recipients), which limits the generalizability of the findings. However, given that the low response rate is a common reality of survey-based research, it is crucial that the authors acknowledge this limitation more transparently and discuss its impact on the reliability of the results. Despite the challenges of achieving a higher response rate, the data collected is commendable for its authenticity and high quality, providing genuine insights into the diversity of practices among respondents.

The authors asked about and obtained key demographic information, such as anesthesiologists' working years, number of beds managed, and nationality, yet these data were not analyzed or discussed in the results. Incorporating subgroup analyses based on these demographic factors would provide valuable insights into how LMA practices vary across regions, experience levels, and healthcare settings. Such analyses could not only elucidate the reasons behind observed heterogeneity but also help tailor recommendations for different settings.

The analysis provided is limited to descriptive statistics, which does not provide sufficient depth for understanding the nuances of LMA use. To strengthen the manuscript, I recommend enhancing the analysis by including comparative statistics that examine differences across various respondent groups, such as geographical differences or variations in practice among anesthesiologists with differing levels of experience. Employing statistical tests to identify significant differences would help strengthen the findings and provide context for the heterogeneity observed.

The presentation of the data also needs improvement. While violin plots are included, they do not adequately convey all aspects of the data. Additional visual tools, such as bar graphs to facilitate categorical comparisons or heatmaps to represent geographical variations, would make the data more comprehensible and actionable for readers.

The discussion attributes the observed heterogeneity in LMA practices primarily to a lack of evidence, but this explanation is overly simplistic. It is essential for the authors to consider additional contributing factors, such as differences in healthcare systems, variations in training, or availability of equipment. Addressing these aspects would lead to a more thorough and balanced discussion of the reasons behind diverse LMA practices.

The conclusion is too generic and lacks specific recommendations. The authors should provide concrete and actionable suggestions for anesthesiologists and outline specific steps that professional bodies could take to develop standardized guidelines. Highlighting the need for prospective randomized studies is important, but practical recommendations based on the survey findings would add significant value to the manuscript.

In summary, while the manuscript addresses an important and relevant topic, it falls short in several critical areas, including insufficient analysis, lack of subgroup comparisons, and vague conclusions that fail to provide actionable insights. I recommend major revisions, with a focus on enhancing methodological rigor, providing in-depth statistical analysis, and including practical recommendations that can be applied in clinical practice.

Reviewer #2: The Authors performed an international survey regarding the use of laryngeal mask airway between anesthesiologists. The survey focused on some of the perceived “limits” in the use of LMA by respondents, such as if anesthesiologists felt there should be a BMI limit, a duration of anesthesia limit, an intraoperative positioning limit, a PEEP o PIP limit and so on. The results are very heterogeneous and in general it is difficult to draw any conclusions other than that there is lack of evidence and lack of agreement between respondents.

The main issues with this work are the following:

- the rate of response was very low, below 5%. Most of the publications regarding methodology of surveys in science papers recommend at least a 30% response rate. This makes interpretation of results very difficult as it is hard to understand how representative are the responses received

- it is not clear what is the ultimate goal of the Authors. A survey is usually the first step of wider research question, useful to gather some preliminary data where there is none or not enough on current practice. This data might be used by the Authors to justify a research proposal, but it is hardly self-sufficient for a publication in the current format

- Figure 1 is only partly available, i.e. only panel A is visible

Reviewer #3: This manuscript brings to light an important topic that is not commonly addressed in airway management about heterogeneity of practices on usage of a laryngeal mask airway (LMA). As highlighted in the manuscript, wide range of many providers don't think about using an LMA or are not comfortable placing an LMA because of a lack of robust evidence or guidelines. I think the appropriate use of an LMA is important to highlight and makes this manuscript worthy of being viewed/read by many. There are, however, some changes to that need to made before I would consider it for publication.

1) I would expand the introduction more. Beyond typographical errors, update the references on recent guidelines and emerging science to highlight current knowledge gaps in airway management and how your work advances this field.

2) I would add more to the limitations. The response rate is low for a study. I would highlight reasons of low survey response and any remediation(s) attempted to overcome them. It is challenging at times to get providers across different settings to complete surveys.

3) Figures don’t appear to display properly in the current version, I see only BMI violin plot and others are not blacked out.

4) Please clarify line 85, PEEP (to answer 16 questions on weight limit, patient positioning, positive).

5) I would expand on the results including details on data on variance (beyond median) as there are concerns regarding the leakage.

6) For the discussion, I would expand more on the interventions to increase the utilization of an LMA. More than a list, I would give some more expansive, specific ideas and then expand into more specific future directions at your institution and beyond.

6. PLOS authors have the option to publish the peer review history of their article (what does this mean? ). If published, this will include your full peer review and any attached files.

**Do you want your identity to be public for this peer review?** For information about this choice, including consent withdrawal, please see our Privacy Policy .

Reviewer #1: **Yes: ** Maohua Wang

Reviewer #2: **Yes: ** Alessandro Santini

Reviewer #3: No

---

## [Author Response · Author response to Decision Letter 0]

19 Feb 2025

We would like to thank you for the reviewers recommendations. We address the comments in the following point-by-point-answers. If appropriate we qoute the changes in the manuscript in the respective answer.

Reviewer #1:

Reviewers remark / question

The response rate of the survey was low (345 out of 7,145 recipients), which limits the generalizability of the findings. However, given that the low response rate is a common reality of survey-based research, it is crucial that the authors acknowledge this limitation more transparently and discuss its impact on the reliability of the results. Despite the challenges of achieving a higher response rate, the data collected is commendable for its authenticity and high quality, providing genuine insights into the diversity of practices among respondents.

Authors answer:

We would like to thank you for this recommendation. We have submitted our survey to the ESAIC, and it was distributed to members via the ESAIC email distribution list. It is customary that, upon explicit request, the ESAIC does not issue a reminder or a second sending of the survey via the ESAIC email distribution list. For this reason, we were unable to enhance the survey's reach and consequently the response rate. However, we agree that the low response rate has to be explained and discussed in more detail. Therefore, the following explanations are included in the paper:

We have added in Methods Section:

“As a rule of ESAIC surveys are sent once without the possibility to send a reminder.”

We have added in Discussion Section:

“Additionally, the low response could have had an impact on the quality of data. Therefore, heterogeneity of the results may be due to the low response rate as well as reflecting reality.”

Reviewers remark / question:

The authors asked about and obtained key demographic information, such as anesthesiologists' working years, number of beds managed, and nationality, yet these data were not analyzed or discussed in the results. Incorporating subgroup analyses based on these demographic factors would provide valuable insights into how LMA practices vary across regions, experience levels, and healthcare settings. Such analyses could not only elucidate the reasons behind observed heterogeneity but also help tailor recommendations for different settings.

The analysis provided is limited to descriptive statistics, which does not provide sufficient depth for understanding the nuances of LMA use. To strengthen the manuscript, I recommend enhancing the analysis by including comparative statistics that examine differences across various respondent groups, such as geographical differences or variations in practice among anesthesiologists with differing levels of experience. Employing statistical tests to identify significant differences would help strengthen the findings and provide context for the heterogeneity observed.

The presentation of the data also needs improvement. While violin plots are included, they do not adequately convey all aspects of the data. Additional visual tools, such as bar graphs to facilitate categorical comparisons or heatmaps to represent geographical variations, would make the data more comprehensible and actionable for readers.

Authors comments:

According to the reviewers recommendation we carried out extensive subgroup analyses on different regions / countries and professional experience and added these results. We present all results of the subgroup analyses in the appendix of this document. Due to the high heterogeneity of participants´ countries and the consecutively very small group sizes for some countries of origin only limited conclusions can be drawn from the results. Although we saw larger groups in terms of professional experience, the analyses carried out did not show any difference that would indicate that professional experience and the use of LMA are related.

To further highlight our results we added boxplots instead of violin plots.

We have added in Methods Section:

“For comparative statistics we used Goodman-Kruskal-Tau-A test for ordinal data and the Cramérs V test for nominal data.”

“Subgroup analyses were performed for duration of professional experience as an anaesthetist ( <6 years ; 6 -10 years ; 11-15 years ; 16-20 years and >20 years ) and the regions of origin of the participants. Goodman-Kruskal-Tau and Cramér's V were calculated to analyse the correlation between professional experience and the limitations of using the LMA. Descriptive data are displayed as boxplots and given as median, quartiles and extremes. Quantitative data for upper limits of PIP, PEEP, BMI and duration were only analyzed when a limit was given by the respondents .”

We have added in the Results section:

“Countries of origin were distributed very heterogeneously. The respondents hailed from 54 countries worldwide, with 31 of these located in Europe, 12 in Africa, 12 in Asia, 4 in America and 2 in Oceania. The detailed data for the countries and regions are listed in the table 1.”

“The subgroup analysis between professional experience and restrictions in the use of the LMA did not reveal any significant association for the correlation in the Goodmann Kruskal tau A test and Cramer's V test.”

Reviewers remark / question

The discussion attributes the observed heterogeneity in LMA practices primarily to a lack of evidence, but this explanation is overly simplistic. It is essential for the authors to consider additional contributing factors, such as differences in healthcare systems, variations in training, or availability of equipment. Addressing these aspects would lead to a more thorough and balanced discussion of the reasons behind diverse LMA practices.

Authors Answer

Thank you for your suggestion to discuss the heterogeneity of the results with regard to the different nationalities and the different experiences of the respondents.

We added in Discussion-Section

“Heterogeneity of LMA use

Our data shows considerable differences in the use of LMA among anaesthetists.

Such variations may arise from differences in healthcare systems, resources and training.

It can be assumed that experienced anaesthetists have more confidence in LMA use and use it for extended indications such as obese patients or prolonged procedures. Inexperienced anaesthetists may act more cautiously, often influenced by theoretical knowledge and safety concerns. However, the respondents‘ level of professional experience was not significantly associated with the surveyed limitations of LMA use. Therefore, we postulate that differences are primarily attributable to uncertainties and a paucity of evidence. However, this reference to the lack of evidence may be too easy. As already discussed, of all the topics addressed in our survey, the safe use of supraglottic airway devices during laparoscopic procedures is the one for which there is the most scientific evidence. However, this question also shows great heterogeneity and a substantial reluctance to use these devices. We can only speculate about the reasons for this but assume that there is insufficient evidence for some questions on the one hand and insufficient acceptance of existing evidence on the other. In short, the traditional strategies from the time when the LMA was developed appear to play a greater role than the available evidence. A major shortcoming in the elective use of LMA is the lack of guidelines. We hypothesise that such a guideline could help to ensure that, in our view, strategies that are no longer up to date are adapted.” 

Reviewer #2:

Reviewers question

The main issues with this work are the following:

a) - the rate of response was very low, below 5%. Most of the publications regarding methodology of surveys in science papers recommend at least a 30% response rate. This makes interpretation of results very difficult as it is hard to understand how representative are the responses received

b) - it is not clear what is the ultimate goal of the Authors. A survey is usually the first step of wider research question, useful to gather some preliminary data where there is none or not enough on current practice. This data might be used by the Authors to justify a research proposal, but it is hardly self-sufficient for a publication in the current format

c) - Figure 1 is only partly available, i.e. only panel A is visible

Authors answer:

a) low response rate

Reviewer 1 made a similar comment, so we take the liberty of repeating the answer here.

Authors answer:

We would like to thank you for this recommendation. We have submitted our survey to the ESAIC, and it was distributed to members via the ESAIC email distribution list. It is customary that, upon explicit request, the ESAIC does not issue a reminder or a second sending of the survey via the ESAIC email distribution list. For this reason, we were unable to enhance the survey's reach and consequently the response rate.

The following explanations are included in the paper:

We have added in Methods Section:

“As a rule of ESAIC surveys are sent once without the possibility to send a reminder.”

We have added in Discussion Section:

“Additionally, the low response could have had an impact on the quality of data. Therefore, heterogeneity of the results may be due to the low response rate as well as reflecting reality.”

b) – ultimate goal:

Authors comment:

The objective of this survey was to establish a database to facilitate a more profound comprehension of contemporary practices and perceptions regarding the utilisation of LMA in extended and borderline indications. We take the liberty of politely disagreeing here. Even if we agree that the results of such a survey cannot be generalised without restriction, they show quite understandably that there are no even reasonably accepted standards for dealing with LMA. We have accepted that the limitations should be emphasised more strongly and have implemented this in the manuscript. Nevertheless, we believe that the results of this survey are worthy of publication in order to emphasize the uncertainties in handling and to demonstrate the very strong need for additional evidence.

To address the Reviewer's concerns, we have added a separate section - Limitations - and drafted it as follows:

“Limitations

An important limitation of this study is the low response rate of less than 5%, which considerably limits the representativeness of the results and may lead to selection bias, as primarily interested or specialised anaesthetists may have participated. Additionally, focusing on ESAIC members may have excluded diverse practices due to differences in healthcare systems, training, and regional standards. The exclusive online format and one-month data collection period might have further limited participation, excluding subgroups. These limitations necessitate caution when interpreting the results and highlight the need for further studies with higher response rates and broader participant bases.”

c)

- Figure 1 is only partly available, i.e. only panel A is visible

Authors answer

We apologize for the incorrect presentation. Following recommendations of the other reviewers we completely reorganised figure 1 a-d in to boxplots. 

we added in the Picture caption:

Fig 1 a-d.

a) max BMI; b) max. duration; c) max. PEEP; d) max. PIP Distribution of responses regarding asked limitations of BMI, duration, PEEP and PIP. Boxplots include median, quartiles, extremes and outliers. BMI (Body Mass Index), PEEP (positive endexpiratory pressure), PIP (peak inspiratory pressure). Data are shown only for respondents who gave an upper limit. 

Reviewer #3:

This manuscript brings to light an important topic that is not commonly addressed in airway management about heterogeneity of practices on usage of a laryngeal mask airway (LMA). As highlighted in the manuscript, wide range of many providers don't think about using an LMA or are not comfortable placing an LMA because of a lack of robust evidence or guidelines. I think the appropriate use of an LMA is important to highlight and makes this manuscript worthy of being viewed/read by many. There are, however, some changes to that need to made before I would consider it for publication.

Reviewers remark / question

1) I would expand the introduction more. Beyond typographical errors, update the references on recent guidelines and emerging science to highlight current knowledge gaps in airway management and how your work advances this field.

Authors Answer

We revised the introduction and have added the following aspects:

1. We referenced existing guidelines for LMA use

2. We discussed uncertainties regarding LMA use in complex clinical scenarios in detail.

3. We emphasized the aim of this international survey

We changed the Introduction Section accordingly:

“Since its introduction in the 1980s, the Laryngeal Mask Airway (LMA) has become a cornerstone of airway management in anaesthesia, offering numerous advantages over endotracheal intubation, including a reduced incidence of complications. Nevertheless, despite its proven benefits, significant uncertainties remain regarding its use in complex clinical scenarios. These include obese patients, cases with increased intra-abdominal pressure, operations exceeding two hours, and specific patient positions like the lateral or prone positions. Furthermore, there is a paucity of evidence regarding the optimal ventilation modes and pressures for LMAs, with these aspects remaining largely unexplored.

This dearth of robust evidence contributes to variability in the practices and perceptions of anaesthetists globally. Guidelines by professional societies these primarily address difficult airway management, rather than the specific limitations of LMA use in the aforementioned scenarios.

The objective of the here presented international survey was to investigate the heterogeneity of LMA usage across diverse clinical environments and scenarios. The identification of existing gaps and regional differences is intended to provide a foundation for multicentre studies that could enhance the evidence base. The survey's findings are expected to underpin the creation of evidence-based guidelines, the harmonisation of standards of care, and the optimisation of LMA use in challenging settings. An international perspective is essential to ensure the safety and efficacy of LMA in complex clinical contexts, thereby fostering innovation and standardisation in airway management.”

Reviewers remark / question

2) I would add more to the limitations. The response rate is low for a study. I would highlight reasons of low survey response and any remediation(s) attempted to overcome them. It is challenging at times to get providers across different settings to complete surveys.

Reviewer 1 and 2 made a similar comment, so we take the liberty of repeating the answer here.

Authors answer:

We would like to thank you for this recommendation. We have submitted our survey to the ESAIC, and it was distributed to members via the ESAIC email distribution list. It is customary that, upon explicit request, the ESAIC does not issue a reminder or a second sending of the survey via the ESAIC email distribution list. For this reason, we were unable to enhance the survey's reach and consequently the response rate.

The following explanations are included in the paper:

We have added in Methods-Section:

“As a rule of ESAIC surveys are sent once without the possibility to send a reminder.”

We have added in Discussion- Section:

“Additionally, the low response could have had an impact on the quality of data. Therefore, heterogeneity of the results may be due to the low response rate as well as reflecting reality.”

We addressed these limitations in the discussion section:

“Limitations

An important limitation of this study is the low response rate of less than 5%, which considerably limits the representativeness of the results and may lead to selection bias, as primarily interested or specialised anesthetists may have participated. Additionally, focusing on ESAIC members may have excluded diverse practices due to differences in healthcare systems, training, and regional standards. The exclusive online format and one-month data collection period might have furthe

---

## [Decision Letter · Decision Letter 1]

7 Mar 2025

Results of an international survey on the use and perceptions of the Laryngeal Mask Airway (LMA)

PONE-D-24-43649R1

Dear Dr. Krug,

We’re pleased to inform you that your manuscript has been judged scientifically suitable for publication and will be formally accepted for publication once it meets all outstanding technical requirements.

Kind regards,

Chinh Quoc Luong, MD., PhD.

Academic Editor

PLOS ONE

Additional Editor Comments (optional):

Reviewers' comments:

Reviewer's Responses to Questions

**Comments to the Author**

1. If the authors have adequately addressed your comments raised in a previous round of review and you feel that this manuscript is now acceptable for publication, you may indicate that here to bypass the “Comments to the Author” section, enter your conflict of interest statement in the “Confidential to Editor” section, and submit your "Accept" recommendation.

Reviewer #2: All comments have been addressed

Reviewer #3: All comments have been addressed

2. Is the manuscript technically sound, and do the data support the conclusions?

Reviewer #2: Yes

Reviewer #3: Yes

3. Has the statistical analysis been performed appropriately and rigorously? 

Reviewer #2: Yes

Reviewer #3: Yes

4. Have the authors made all data underlying the findings in their manuscript fully available?

Reviewer #2: No

Reviewer #3: Yes

5. Is the manuscript presented in an intelligible fashion and written in standard English?

Reviewer #2: Yes

Reviewer #3: Yes

6. Review Comments to the Author

Reviewer #2: (No Response)

Reviewer #3: Dear authors, thank you for giving me the opportunity to review your manuscript and incorporating some of my suggestions. I have nothing to recommend to revise.

7. PLOS authors have the option to publish the peer review history of their article (what does this mean? ). If published, this will include your full peer review and any attached files.

**Do you want your identity to be public for this peer review?** For information about this choice, including consent withdrawal, please see our Privacy Policy .

Reviewer #2: No

Reviewer #3: No

---

## [Editor Report · Acceptance letter]

PONE-D-24-43649R1

PLOS ONE

Dear Dr. Krug,

I'm pleased to inform you that your manuscript has been deemed suitable for publication in PLOS ONE. Congratulations! Your manuscript is now being handed over to our production team.

Kind regards,

on behalf of

Assoc. Prof. Chinh Quoc Luong

Academic Editor

PLOS ONE